# Therapeutic Potential of Small Molecules Targeting Oxidative Stress in the Treatment of Chronic Obstructive Pulmonary Disease (COPD): A Comprehensive Review

**DOI:** 10.3390/molecules27175542

**Published:** 2022-08-28

**Authors:** Hamad Ghaleb Dailah

**Affiliations:** Research and Scientific Studies Unit, College of Nursing, Jazan University, Jazan 45142, Saudi Arabia; hdaelh@jazanu.edu.sa; Tel.: +966-551997991

**Keywords:** chronic obstructive pulmonary disease (COPD), small molecules, reactive nitrogen species, reactive oxygen species, oxidative stress, cigarette smoke, antioxidants

## Abstract

Chronic obstructive pulmonary disease (COPD) is an increasing and major global health problem. COPD is also the third leading cause of death worldwide. Oxidative stress (OS) takes place when various reactive species and free radicals swamp the availability of antioxidants. Reactive nitrogen species, reactive oxygen species (ROS), and their counterpart antioxidants are important for host defense and physiological signaling pathways, and the development and progression of inflammation. During the disturbance of their normal steady states, imbalances between antioxidants and oxidants might induce pathological mechanisms that can further result in many non-respiratory and respiratory diseases including COPD. ROS might be either endogenously produced in response to various infectious pathogens including fungi, viruses, or bacteria, or exogenously generated from several inhaled particulate or gaseous agents including some occupational dust, cigarette smoke (CS), and air pollutants. Therefore, targeting systemic and local OS with therapeutic agents such as small molecules that can increase endogenous antioxidants or regulate the redox/antioxidants system can be an effective approach in treating COPD. Various thiol-based antioxidants including fudosteine, erdosteine, carbocysteine, and N-acetyl-L-cysteine have the capacity to increase thiol content in the lungs. Many synthetic molecules including inhibitors/blockers of protein carbonylation and lipid peroxidation, catalytic antioxidants including superoxide dismutase mimetics, and spin trapping agents can effectively modulate CS-induced OS and its resulting cellular alterations. Several clinical and pre-clinical studies have demonstrated that these antioxidants have the capacity to decrease OS and affect the expressions of several pro-inflammatory genes and genes that are involved with redox and glutathione biosynthesis. In this article, we have summarized the role of OS in COPD pathogenesis. Furthermore, we have particularly focused on the therapeutic potential of numerous chemicals, particularly antioxidants in the treatment of COPD.

## 1. Introduction

Chronic obstructive pulmonary disease (COPD) is a chronic respiratory disease. One of the very common characteristics of COPD includes irreversible obstructive breathing. The World Health Organization (WHO) identified COPD as the third leading cause of death worldwide in 2019 [1]. Oxidative stress (OS) has significant effects on various lung functions and in COPD pathogenesis. These effects include apoptosis, remodeling of the extracellular matrix, alveolar epithelial injury, mitochondrial respiration, membrane lipid peroxidation (LPO), mucus hypersecretion, and oxidative inactivation of surfactants and antiproteases [2,3]. In the respiratory tracts of COPD patients, an elevated level of OS is observed because of the elevated oxidant burden from environmental exposure including air pollutants and cigarette smoke (CS), and from the elevated levels of reactive nitrogen species (RNS) and reactive oxygen species (ROS) secreted from the macrophages and leukocytes associated with the inflammatory processes in the lungs of COPD patients [4,5,6]. These RNS and ROS have the capacity to cause oxidative damage to proteins, carbohydrates, lipids, and DNA, which can further lead to various downstream mechanisms that can lead to COPD development and progression. In addition, they can also cause the activation of resident cells in the lung including alveolar macrophages and epithelial cells to produce chemotactic molecules that engage more inflammatory cells (such as lymphocytes, monocytes, and neutrophils) into the lung [4,7,8], which eventually result in the spread of OS in the lung. As a whole, these processes result in a dangerous cycle of tenacious inflammation along with chronic OS, which can result in faulty tissue repair processes, disruptions in the protease–antiprotease balance, increased autophagy, and apoptosis, which can eventually play roles in the progression and severity of COPD [9,10,11,12].

ROS is the main cause of tissue and cell injury linked with numerous inflammatory pulmonary diseases including COPD [13,14]. Nonetheless, the exact mechanisms of COPD pathogenesis are yet to be fully revealed. Oxidant/antioxidant imbalance and elevated levels of OS may be the main causes of COPD. An elevated level of OS is produced from airway leukocytes in the blood or in air spaces indirectly because of the secretion of elevated levels of ROS, and directly because of the environmental oxidant pollutants and CS [15]. It is well-known that antioxidant enzymes and compounds scavenge ROS [13,16]. Cigarette smoking is the major etiological factor in COPD pathogenesis, since it can result in OS in the lower airways. Furthermore, CS contains over 1016–1017 oxidants per puff and around 4700 chemicals such as nitrogen oxides, superoxide radicals, and peroxynitrite (ONOO^−^) [17]. CS exerts significant adverse effects; nonetheless the effects of other risk factors also need to be well-considered, since not all smokers develop COPD. In Table 1, we summarized the risk factors that are linked with COPD development. Multiple intracellular and extracellular antioxidants provide protection to the blood and lungs against the harmful activities of oxidants, under normal conditions [15]. It has been reported that antioxidants including thiol compounds/donors and their analogs including glutathione (GSH) and mucolytic drugs, for example, fudosteine, erdosteine, carbocysteine, and N-acetyl-L-cysteine have the ability to efficiently regulate nuclear factor-κB (NF-κB) activation, elevate intracellular thiol concentrations, and detoxify/scavenge oxidants/radicals, therefore suppressing inflammatory gene expressions [18]. In this article, we have highlighted the important role of OS in COPD pathogenesis. We have also particularly focused on the therapeutic potential of numerous antioxidants in the treatment of COPD.

## 2. Reactive Oxygen Species

OS is supposed to be triggered via CS, and OS-mediated cell injury has a significant contribution in the development of COPD [47]. CS contains a complex combination of many free radicals and ROS, which is further grouped into two phases including tar and gas. The tar phase contains around 1017 long-lived free radicals per gram, for instance, quinone/hydroquinone (Q/QH2) radicals can decrease oxygen to generate superoxide anions (O_2_^•−^), resulting in the production of hydroxyl radicals (^•^OH) (Table 2) and hydrogen peroxide (H_2_O_2_) [48,49]. It has been reported that ^•^OH is a highly reactive ROS, which can injure nearly all types of macromolecules following collision [50]. Interestingly, hydroxyl radicals can be produced via a Fenton reaction including H_2_O_2_ and cuprous copper (Cu(I)) or ferrous iron (Fe(II)), which comprise harmful connections of redox- and metal homeostasis. Particulate matter (also known as particle pollution) increases OS, which can further damage the Fenton reaction and lead to the generation of ^•^OH in the lung [51]. Superoxide radical anions (O_2_^•−^) are less reactive compared to ^•^OH, however, O_2_^•−^ is also detrimental and can play a role in the one-electron pathway involving flavin cofactors and metals. In contrast, H_2_O_2_ is comparatively stable, which has the capacity to travel long distances from its location of generation [50].

Compared to the superoxide radical anions, H_2_O_2_ mainly participates in two-electron pathways involving sulfur-containing moieties in the cell. Nonetheless, H_2_O_2_ also plays a role in some one-electron pathways involving transition metals. Interestingly, H_2_O_2_ can play a role as a damaging agent at higher concentrations and as a signaling molecule at low concentrations. Therefore, H_2_O_2_ exerts a significant cellular effect that is defined via overlapping processes of H_2_O_2_ identification, signal transduction, and obliteration [52,53]. Hypochlorous acid (HOCl) produced in the presence of H_2_O_2_ can result in the generation of more harmful ROS including ^•^OH [54]. Hypochlorite anions (^−^OCl) exhibit increased reactivity, which suggests that it indiscriminately modifies its targets, usually with second-order rate constants of 10^5^–10^7^  M^−1^ s^−1^ [55]. In the case of proteins, various amino acids including methionine, histidine, and cysteine are known as the ideal residues for modification. Moreover, ^−^OCl has the capacity to alter various primary amines (those are present in the lysine’s sidechain) to chloramines. Collectively, increased concentrations of ROS might result in respiratory problems and lung tissue injury by modifying several target molecules by different ROS-specific processes.

On the other hand, the gas phase of CS possesses an increased level of reactive molecules compared to tar. In addition, the gas phase possesses 1015 inorganic and organic radicals per puff including ONOO^−^, nitrogen dioxide, and nitric oxide (NO^•^) [49,56]. NO^•^ is one of the major RNS and ROS. CS also possesses around 74.5–1008 ppm NO^•^ and smokers are mostly exposed to this free radical. Although NO^•^ has a shorter half-life (t_1/2_= around 0.09–2 s), this free radical reacts rapidly with O_2_^•−^ to generate ONOO− [49,57,58,59]. ONOO^−^ (another RNS) is associated with numerous pathological and physiological mechanisms [60,61]. ONOO^−^ also has powerful nitration and oxidation capacities, which can further damage various molecules in the cells including proteins and DNA. Interestingly, a second-order reaction relies on the levels of two first-order reactants or one second-order reactant, which are NO^•^ and O_2_^•−^ in the generation of ONOO^−^. NO^•^ also has the capacity to interact with organic lipid peroxyl radicals (ROO^•^) (that are found in CS) to generate cytotoxic species including alkyl peroxynitrites (ROONO). It has been observed that O_2_^•−^ and NO^•^ are generated via multiple inflammatory cells including macrophages via nicotinamide adenine dinucleotide phosphate (NADPH) oxidase complexes (NOXs) and nitric oxide synthases (NOSs), respectively. In the endoplasmic reticulum, RNS and ROS can also be secreted via an uncontrolled mechanism as by-products during various processes including the protein folding maturation process, peroxisomal metabolism, and mitochondrial respiration [62,63]. Moreover, their elevated generation can eventually result in OS and lung injury.

**Table 2 molecules-27-05542-t002:** A summary of the free radicals that can play a role in oxidative stress.

Name	Structural Formula	Properties	References
**Reactive oxygen species**			
**Radicals**			
Hydroxyl radical	^•^OH	Highly reactive, very unstable in aqueous solutions	[64]
Superoxide	^•^O_2_^−^	Moderately reactive, highly unstable, modulate signaling	[65,66]
Peroxyl radical	ROO^•^	Products of lipid peroxidation	[67]
Alkoxyl radical	RO^•^	Products of lipid peroxidation	[67]
**Non Radicals**			
Hydrogen peroxide	H_2_O_2_	Toxic, associated with several signal transduction pathways and cell fate decisions	[68,69]
Hypochlorite anion	OCl^−^	Produced by myeloperoxidase	[70]
Singlet oxygen	^1^O_2_	highly excited, nonradical, metastable state of molecular oxygen	[67]
Ozone	O_3_	Environmental air pollutant	[71]
**Reactive nitrogen species**			
**Radicals**			
Nitrogen dioxide	^•^NO_2_	One of the most threatening environmental air pollutants, highly reactive	[65]
Nitric oxide	^•^NO	Important redox signaling molecule	[72]
**Non Radicals**			
Nitrogen oxides	NO_x_	Environmental toxins including NO and ^•^NO_2_ linked with combustion sources	[65,73]
Peroxynitrite	ONOO^−^	Highly reactive, unstable intermediate	[74,75]

## 3. Endogenous and Exogenous Generations of Reactive Oxygen Species

The lungs can be exposed to ROS and RNS derived from environmental and cellular sources. CS is the major environmentally derived ROS that induces COPD pathogenesis. RNS/ROS can also be produced via multiple structural and inflammatory cells of the airways. The inflammatory response (inflammation) is a feature of COPD, which is characterized via activation of resident macrophages and epithelial cells, and the recruitment and activation of monocytes, neutrophils, and T- and B-lymphocytes. Once employed in the airspace, inflammatory cells become activated and produce ROS as a reaction to an adequate concentration of a secretagogue stimulus (threshold condition). In inflammatory cells, NADPH oxidase (NOX) is the major ROS-generating enzyme. Various other enzyme systems including the heme peroxidases and xanthine/xanthine oxidase (XO) system are also associated with COPD [76,77]. In a similar manner, nitric oxide synthase mediates the generation of RNS in the form of nitric oxide (NO) generation. In the presence of superoxide anions, nitric oxide generates more strong and harmful ONOO^−^ molecules. It has been reported that the disturbance of the components of NOX including gp91^phox^ and p47^phox^, which was found to enlarge the airspace in mouse models, further indicating that ROS-derived from NOX can play a role in the signaling pathways of tissue homeostasis. Moreover, in COPD, the use of inhibitors of NOX to redress the imbalance between antioxidants and oxidants might be harmful [78].

## 4. Sources of Reactive Oxygen Species in the Lung

The lung is susceptible to damage from environmental OS due to its anatomic structure. In addition, the lung is also exposed to various sources of endogenous OS produced via mitochondrial respiration and inflammation to viral and bacterial infections. The environmental sources of airborne OS involve oxidant gases and nanoparticles and ultrafine particles from car exhaust fumes and industrial pollution. Nonetheless, tobacco smoking is the most significant risk factor for COPD in the Western world. On the other hand, the inhalation of combustion products from enclosed cooking fires is a significant additional etiological factor in developing countries [79]. Although tobacco smoking can trigger the onset of COPD (Figure 1), once COPD has occurred, smoking cessation does not stop COPD progression and the continuous presence of OS [80]. Various endogenous sources including mitochondrial respiration mediate the continuous presence of OS. In the presence of carbonyl stress, airway epithelial cells trigger the generation of ROS derived from mitochondria. In COPD patients, airway smooth muscle cells generate an increased level of mitochondrial-derived ROS when exposed to inflammatory stress from interferon-gamma, tumor necrosis factor-alpha, and interleukin (IL)-1. It has been revealed by pathway studies that mitochondrial dysfunction around complexes I and III are closely linked with COPD (Figure 1) [81,82]. Furthermore, various other intracellular sources of ROS include several cytoplasmic ROS-generating enzymes including heme peroxidases, xanthine/XO system, and NOX. The levels of these enzymes were found to be increased in inflammatory cells within the airways of COPD patients [83]. Although profusely generated superoxide radicals are a comparatively weaker oxidizing agent, however, it is the precursor for various other more detrimental ROS species including hydroxyl radicals, which are increased in the case of COPD, or the highly damaging and strong ONOO^−^ radicals generated by the fast reaction of superoxide with nitric oxide [3,84,85]. In a similar manner, myeloperoxidase (MPO), secreted by activated neutrophils, is likely to build up in the lungs of COPD patients, which can further result in the generation of highly damaging HOCl. Nevertheless, intracellular antioxidant defenses are able to effectively clean up these ROS species in healthy cells, and can therefore limit their impact.

## 5. Role of Oxidative Stress in COPD Pathogenesis

A comparative deficit of antiproteases including alpha-1 antitrypsin (AAT) (because of their inactivation mediated via oxidants) may trigger the imbalance between protease and antiprotease in the lungs, which can further form the basis of the protease–antiprotease paradigm of emphysema pathogenesis [86]. Various studies have revealed the in vitro AAT inactivation via oxidants secreted from inflammatory leukocytes or oxidants from CS, but this activity is less common in vivo [87]. A protease–antiprotease imbalance including elastase and AAT is an overgeneralization, as other antiproteases and proteases are also likely to be involved. It has been reported that oxidant generating systems including xanthine/XO can induce the secretion of mucus from respiratory epithelium [88,89]. In addition, oxidants are associated with the signaling mechanisms for the epidermal growth factor receptor (EGFR), and EGFR plays a vital role in mucus secretion [90]. At low concentrations (100 μM), oxidants including HOCl or H_2_O_2_ can cause marked damage to ciliary beating and stasis [89,91]. Therefore, oxidant-induced hypersecretion of mucus and reduced mucociliary clearance might lead to mucus accumulation in the airways, which can eventually result in airflow limitation. Damage to the respiratory epithelium is a key early event that takes place following exposure to CS, which can be detected by an elevated level of the epithelial permeability of the airspaces. Both in vitro and in vivo studies have revealed that exposure to CS can lead to elevated epithelial permeability, which is partly reversible by antioxidants [92,93,94]. Intra and extracellular GSH seems to be essential in maintaining epithelial integrity following exposure to CS, as deficiency of lung GSH alone may result in an elevated level of in vitro and in vivo epithelial permeability of the airspaces [95,96]. Interindividual differences in antioxidant defenses might be a factor in defining whether COPD occurs in cigarette smokers.

It has been confirmed by biopsy studies that COPD is linked with an increased level of inflammation in the airways [97]. OS might be a process through which the inflammation of airspaces is increased in the case of COPD [98]. OS might also have a significant contribution in increasing inflammation via upregulating various redox-sensitive transcription factors including activator protein-1 (AP-1) and NF-κB. Furthermore, OS might also increase inflammation via upregulating various extracellular signal-regulated kinases including p38 mitogen-activated protein-kinase and c-Jun-N-terminal kinase signaling mechanisms. It has been reported that CS activates all of these aforesaid signaling mechanisms [99,100]. Genes for numerous inflammatory mediators are controlled via various oxidant-sensitive transcription factors including NF-κB [101,102,103]. Oxidants caused the secretion of multiple inflammatory mediators including NO, IL-1, and IL-8 in bronchial epithelial cells, alveolar epithelial cells, and macrophage cell lines, and these processes were found to be linked with an elevated level of expressions of the genes for these inflammatory mediators, and an elevated level of activation and the nuclear binding of NF-κB [104,105]. It has been observed that NF-κB can be activated and translocated to the nucleus in the lung tissues in COPD patients and cigarette smokers compared to healthy individuals [99,106,107].

Activation of NF-κB in lung tissues is associated with the forced expiratory volume in the first second (FEV1) [108]. Interestingly NF-κB linking to its consensus location in the nucleus can result in increased pro-inflammatory gene transcriptions, and as a result inflammation, which can further lead to the increased OS and can create a vicious circle of elevated levels of inflammation resulting from the elevated level of OS (Figure 2). The influx of neutrophils in the lungs was found to be linked with an elevated level of NF-κB activation and the expressions of IL-8 genes and protein secretion in animal models of smoke exposure [109]. Indeed, all of these aforementioned processes were reported to be linked with OS, as these processes can be abrogated via antioxidant therapy [110,111,112]. OS can also induce chromatin remodeling, which might increase the level of inflammation in the lungs, further permitting access for RNA polymerase and NF-κB to the transcriptional machinery to increase the gene expression, and this event is familiar as oxidant sensitive [98]. It is hypothesized in emphysema pathogenesis that the loss of lung cells (particularly endothelial cells) might take place as a primary mechanism in the emphysema development due to the apoptosis of endothelial cells [113]. An increased level of apoptosis has been observed in emphysematous lungs compared to the lungs of nonsmokers [114]. Vascular endothelial growth factor receptor 2 (VEGFR2) can also mediate endothelial apoptosis. VEGFR2 downregulation resulted in emphysema in animal models and such VEGFR2 downregulation has also been detected in emphysematous lungs [114]. Apoptosis/emphysema mediated by the suppression of VEGF in animal models was linked with elevated OS markers and was averted via antioxidants [113], further indicating that OS is associated with this event and systemic effects including muscle dysfunction in the case of COPD [115].

## 6. Pulmonary Strategies of Antioxidant Defense

### 6.1. Glutathione

Since the lungs are continuously exposed to endogenous and external sources of OS, they have created various effective antioxidant defensive approaches (Figure 1), where a decreased GSH level has a significant contribution. As a by-product of metabolism, as much as 20% of all generated GSH can be found within the mitochondria to counteract the production of endogenous ROS [116]. It has been revealed that the introduction of airway epithelial cells from healthy individuals to acute OS induces an elevated level of GSH synthesis via increasing the expression and function of glutamyl-cysteine ligase (an important enzyme in the synthesis of GSH) [117]. Nevertheless, the level of glutamyl-cysteine ligase decreases near the central bronchial epithelium and in alveolar macrophages in COPD patients and smokers, which indicates a faulty regulatory process [118]. Comparable differential reactions between the control individuals and COPD patients were obvious with various other GSH-dependent antioxidant enzymes including the glutathione-S-transferase (GST) pi isoenzyme, GSH peroxidase, and GSTM1 [119]. Moreover, a genetic deletion mutation in GSTM1 was liked with emphysema development in smokers and elevated vulnerability in COPD development [120]. In a similar manner, COPD has also been linked with the genetic polymorphisms in the GST pi isoenzymes [121]. 

### 6.2. Restoration or Upregulation of Nrf2 Function

In the airway smooth muscle cells, decreased levels in the expressions of various antioxidant enzymes including superoxide dismutase 2 (SOD2) and catalase were decreased, while the expression of transforming growth factor β (TGF-β) was elevated in the case of COPD [122]. These aforesaid antioxidant enzymes are important for counteracting mitochondrial-derived ROS and are controlled by the transcription factor forkhead box O-3 (FoxO3). It has been observed that there is a deficit in FOXO3 function in the case of COPD [123]. SOD2 gene polymorphism is also strongly linked with COPD; however, insufficient data are available to demonstrate how these polymorphisms are associated with the functions [124]. SOD3 polymorphisms have also been linked with decreased lung activity in the case of COPD and defense against COPD development in smokers when activities of SOD3 are increased [125,126]. Nuclear erythroid-2-related factor 2 (Nrf2) controls more than 200 cellular detoxification and antioxidant enzymes, which also control the expression of genes via binding with the antioxidant response elements within the promoters of numerous cytoprotective and antioxidant genes [127]. A decreased level of expressions of Nrf2-responsive genes because of decreased Nrf2 function has been reported in COPD patients [128]. Thus, the restoration or upregulation of Nrf2 function might be an effective therapeutic approach in the case of COPD [129].

## 7. Oxidative Stress Biomarkers in Chronic Obstructive Pulmonary Disease

### 7.1. Exhalation of NO^•^

In the lungs, the gas NO^•^ is endogenously generated via NO synthase (NOS), which can exist in both the inducible isoform (iNOS) and constitutive isoform (cNOS). iNOS can be mediated via inflammatory stimuli in the lungs and might thus indicate airway inflammation. Thus, exhaled NO^•^ levels are regarded as an indirect measure of OS and a marker for airway inflammation [130,131]. Nonetheless, the findings on the usage of NO^•^ concentrations as a marker for COPD are questionable. In stable COPD patients, an increased level of NO^•^ was found in exhaled air whereas others had lower or normal levels of exhaled NO^•^ compared to the control individuals [132]. These inconsistencies in the findings might take place because of the different criteria for patient selection or the use of different approaches of measurement. In addition, NO^•^ is short-lived in vivo and by rapidly reacting with the superoxide, it can be transformed into NOx [133,134]. NO^•^ might also produce stable S-nitrothiols (RS-NOs) by interacting with low molecular weight thiols including N-acetylcysteine or GSH to increase its bioactivity [135,136,137]. Therefore, instead of NO^•^, RS-NOs are considered as the major products of inflammation and NOS. Compared to the healthy control individuals and nonsmokers, reports on the RS-NOs concentrations in inflammatory airway diseases exhibited elevated concentrations in the exhaled breath condensate (EBC) of smokers and patients with COPD [130].

### 7.2. Exhalation of Hydrogen Peroxide

H_2_O_2_ in exhaled air signifies oxidant production in the lungs. It has been reported that COPD patients and smokers exhale H_2_O_2_ much higher than nonsmokers or ex-smokers with COPD [138,139]. The level of H_2_O_2_ in exhaled air is even higher during acute exacerbations of COPD. Although the exact cause of this increased exhalation of H_2_O_2_ is not fully known, this may partially take place from the elevated superoxide anion (O_2_^−^) secretion by the alveolar macrophages from smokers than the alveolar macrophages from nonsmokers [140,141]. Compared to nonsmoking individuals, an increased level of intracellular iron has been observed in the pulmonary macrophages of smokers [142]. In the airspaces of smokers, the presence of elevated levels of free irons might elevate the production of even more ROS via the Fenton reaction [143]. The combination of xanthine/XO has the capacity to produce H_2_O_2_ and superoxide anion radicals. This combination was found to be elevated in the plasma and bronchoalveolar lavage (BAL) of smokers and COPD patients compared to nonsmokers and healthy individuals, respectively. Interestingly, patients with COPD performing strenuous exercise experienced systemic OS, which was suppressed via blocking XO [4].

### 7.3. Inflammatory Response

Multiple studies have assessed the contribution of ROS in the production of the inflammation that takes place in the peripheral and central airways of patients with COPD [100,144]. Common characteristics of lung inflammation include the activation and recruitment of neutrophils, and the activation of resident macrophages and epithelial cells [100]. Oxidants that are present in CS have the capacity of inducing alveolar macrophages to secrete various mediators, some of which engage neutrophils and various other inflammatory cells in the lungs [100,145,146,147,148]. In addition, elevated numbers of neutrophils and macrophages move into the lungs of smokers, where these cells produce ROS by the reduced NOX [149,150,151,152,153]. The lungs of smokers containing airway obstructions possess more neutrophils compared to smokers without airway obstruction [154,155]. During acute exacerbations, peripheral blood neutrophils from COPD individuals and smokers exhibit an elevated generation of superoxide anions. In patients with COPD, this generation level was found to be returned to normal when they were clinically stable [156,157]. It has been reported that the MPO level of the neutrophils is positively linked with tobacco smoking, indicating an elevated level of the generation of oxidants including hypochlorous acid in smoking individuals [83]. A connection between the FEV1 and circulating neutrophil numbers has also been observed, which further indicates an elevated level of airflow limitation due to the ROS generation of the elevated level of neutrophils [158]. An elevated level of ROS release takes place from the circulating neutrophils in tobacco smokers with COPD compared to tobacco smokers without COPD [159].

### 7.4. Lipid Peroxidation (LPO)

In biological tissues, ROS can induce the peroxidation of polyunsaturated fatty acids, which can lead to the transformation of fatty acids into lipid hydroperoxides. Then, lipid hydroperoxides and lipid peroxides can react with nonenzymatic or enzymatic antioxidants or decay after interacting with iron-containing proteins or metal ions, which can lead to the formation of unsaturated aldehydes and hydrocarbon gases as by-products [160,161]. Lipid peroxidation (LPO) measured as thiobarbituric acid-reactive substances showed an increase in concentrations in the lungs and breath condensate of stable COPD individuals [141,162]. Furthermore, these LPO products are negatively linked with the FEV1, which indicates that LPO has a significant contribution in deteriorating lung activity [163]. An increased level in the LPO product has been observed in the bronchoalveolar lavages and plasma of healthy smokers. Moreover, elevated LPO product levels are inversely linked to the extent of small airway obstruction and time expired from the last exposure to CS [164,165]. 4-Hydroxy-2,3-nonenal (an end-product of LPO) has the capacity to modify cellular proteins. The respiratory endothelial and epithelial cells of smokers with airway obstructions had increased levels of 4-hydroxy-2,3-nonenal-modified proteins in comparison with the individuals without airway obstruction or nonsmokers [166]. Ethane (a hydrocarbon) can be generated as a by-product of the peroxidation of various fatty acids including 9,12,15-linolenic acid [131]. Exhaled ethane is increased in the case of COPD patients compared to the control individuals. This elevated level is negatively associated with lung activity, which indicates that LPO is a crucial factor in COPD progression [131].

### 7.5. Protein Degradation

OS makes proteins more vulnerable to proteolytic degradation by altering amino acid chains, which results in the formation of protein aggregates and the cleavage of peptide bonds. Some of the amino acid residues are transformed to carbonyl residues during this process, which can be found systemically [167,168,169,170]. Following exposure to the gas phase of CS, human plasma proteins are altered to carbonyl-containing proteins with lost sulfhydryl groups. In CS, the unsaturated and saturated aldehydes play roles in modifying proteins [171,172]. It has been revealed by in vitro studies that the exposure of human plasma to CS resulted in raised levels of carbonyl proteins and depleted levels of plasma protein sulfhydryls [173]. OS-mediated damage of proteins and thus the generation of carbonyl proteins due to the exposure of CS were found to be partially prevented by GSH and completely by ascorbic acid [174]. Cigarette smoking can also lead to the formation of RNS, which can also cause the degradation of plasma proteins via oxidation and nitration [100]. Compared to nonsmokers, concentrations of oxidized proteins are considerably higher in smokers [175], and compared to nonsmokers, smoking individuals also showed increased levels of various nitrated proteins including plasminogen, ceruloplasmin, transferrin, and fibrinogen [175].

Aldehydes found in CS might interact with the amino and sulfhydryl moieties of plasma proteins through a Michael reaction [176,177]. The transformation of tyrosine into dityrosine and 3-nitrotyrosine can be considered as an indicator of protein and free radical damage [178,179]. The levels of nitrotyrosine were found to be increased in the epithelial lining fluid and plasma of tobacco smokers and negatively linked with the FEV_1_ [180]. Alpha 1-proteinase inhibitor (alpha 1-PI), an inhibitor of elastase activity, has a significant contribution in patients with COPD and its activity was reported to be reduced by oxidizing agents [181,182]. In alpha 1-PI, the oxidation of an important residue of methionine amino acid into methionine sulfoxide markedly reduced the suppressive ability of alpha 1-PI [170,183,184]. It has been revealed by BAL that smokers contain alpha 1-PI, which only exhibited half of its normal function, whereas alpha 1-PI from the lung washings of nonsmoking individuals is fully functional with only native methionine [141].

## 8. Strategies for Reducing Oxidative Stress by Antioxidants in the Treatment of COPD

### 8.1. Thiol-Based Antioxidants

#### 8.1.1. Carbocysteine

S-carboxymethylcysteine (S-CMC or carbocysteine) (Figure 3) is a thiol derivative of L-cysteine. It has been revealed that S-CMC has anti-inflammatory, mucoactive, and antioxidant effects. Currently available oral preparations of carbocysteine include S-CMC and lysine salt of S-CMC (S-CMC-lys). In the gastrointestinal (GI) tract, the lysine residue in S-CMC-Lys breaks down to generate the active drug S-CMC. Compared to erdosteine and N-acetyl-L-cysteine (NAC), the mucoactive effect of carbocysteine is different from other thiol-based mucolytics as carbocysteine elevates the sialomucin level, which affects the rheological effects of mucus through the suppression of kinins [185]. Furthermore, carbocysteine mediates mucociliary clearance velocities, specifically in individuals with chronic bronchitis who have a lower level of clearance prior to treatment [185]. Carbocysteine also provided protection against emphysema triggered by CS in the rat models [186]. In patients with COPD, the treatment with S-CMC-Lys for a duration of 6 months markedly reduced the concentrations of the 8-isoprostane (an LPO product) and pro-inflammatory cytokine including IL-6, which suggests that S-CMC-Lys showed both antioxidant and anti-inflammatory effects [187]. Since this drug can also decrease the levels of bacterial respiratory tract infections in the case of COPD [47], carbocysteine might therefore play a role by suppressing the adherence of pathogens to the cells.

Compared to the placebo-treated group, the in vitro studies revealed that treatment with carbocysteine can decrease the attachment of *Moraxella catarrhalis* (a bacterium to pharyngeal epithelial cells in individuals with chronic bronchitis and healthy subjects [188,189,190]. Carbocysteine can also markedly decrease the adherence of *Streptococcus pneumoniae* to pharyngeal epithelial cells [47]. In COPD patients, carbocysteine can decrease the occurrence of common colds and related exacerbations, a characteristic that can contribute to its capacity to reduce the expression of intercellular adhesion molecule 1 in the respiratory tract [191]. The effect of treatment with carbocysteine (250 mg, 3 times a day) has been assessed for 3 years in 709 Chinese COPD patients. It was observed in a clinical study that COPD patients receiving carbocysteine faced less frequent exacerbations per year [192]. Moreover, most of these COPD patients were not administered corticosteroids [192]. The clinical trials of carbocysteine in COPD patients are summarized in Table 3.

#### 8.1.2. N-Acetyl-L-Cysteine

NAC (Figure 3) is a powerful reducing agent and an acetyl derivative of the amino acid. NAC has mucolytic properties and it decreases the viscosity of mucus, thus ameliorating mucociliary clearance. On the other hand, in the GI tract, NAC is deacetylated to cysteine, which plays a role as a GSH precursor. NAC has the capacity to neutralize oxidant species by reducing disulfide bonds. In lungs, NAC can also elevate the level of intracellular GSH in vivo as it can reduce intracellular cystine to cysteine. Among the thiol derivatives, NAC has been extensively studied both in vivo and in vitro. Oral administration of NAC decreased the level of elastase-induced emphysema in rat models [198]. Furthermore, NAC provided protection against the tobacco smoke-mediated oxidation of the Z variant of AAT in early-onset emphysema in mouse models [199]. Since GSH is an important lung antioxidant, NAC has primarily been utilized to increase the level of lung GSH in COPD patients [200,201]. In Table 4, the clinical trials on the benefits of using NAC in COPD patients have been summarized. 

In a clinical study, NAC treatment at the dose of 600 mg twice per day for 6 months decreased several BAL fluid and plasma oxidative biomarkers in tobacco smokers [210]. On the other hand, NAC treatment at the dose of 600 mg two times per day for 2 months decreased the oxidative burden in the airways of individuals with stable COPD [207], and was linked with ameliorated lung symptoms and decreased risk of exacerbation in people with chronic bronchitis [206]. In severe COPD patients, NAC ameliorated the muscle activity by increasing the time of quadriceps endurance, along with a reduction in the systemic OS markers [211]. Various reports have also shown reduced levels of exacerbations by 29% [212,213]. In a large multicenter trial, NAC did not affect the decline in FEV1 or the frequency of the exacerbations [203]. Moreover, in this study, a reduced level of exacerbation frequency and overinflation was observed in COPD patients who did not receive inhaled glucocorticoids [203]. In the GI tract, NAC needs to be deacetylated to cysteine to play a role as a GSH precursor and is not highly bioavailable to elevate the level of GSH. Therefore, in order to identify any clinical advantage in COPD treatment, more studies are needed to evaluate the effects of NAC at higher doses (1200 mg or 1800 mg daily) or evaluate various other thiol derivatives that show higher bioavailability.

#### 8.1.3. Fudosteine

Fudosteine (Figure 3) is used as an antioxidant and mucolytic agent. Compared to NAC, fudosteine has higher bioavailability and plays a role as an antioxidant by elevating the level of intracellular cysteine. Fudosteine also suppressed the hypersecretion of mucin by downregulating the gene expression of Mucin 5AC (MUC5AC) [214]. Fudosteine decreased the expressions of p-ERK in in vitro bronchial epithelial cell lines and in vivo expressions of p-ERK and phospho-p38 MAPK [214]. In lung epithelial cells, fudosteine suppressed the airway nitrative stress induced by ONOO^−^ by directly scavenging ONOO^−^ [215]. Therefore, as a mucoactive agent, fudosteine might be useful in the therapy of various chronic respiratory diseases such as bronchiectasis, COPD, chronic bronchitis, and bronchial asthma [214,216].

#### 8.1.4. Erdosteine

Erdosteine (Figure 3) is a thiol-based antioxidant and mucoactive agent. Erdosteine was mainly utilized as a mucolytic agent. It breaks the disulfide bonds of mucus glycoproteins and affects the physical properties of the mucus, which can further result in an elevated level of mucus clearance [217]. In addition, erdosteine has antibacterial, anti-inflammatory, and antioxidant properties. In a clinical trial, the oral administration of erdosteine (300 mg twice daily) for 8 months markedly reduced exacerbations and improved the quality of health in comparison with the placebo [218]. Erdosteine can also be beneficial for people suffering from severe, prolonged, or repeated COPD exacerbations [219,220]. Erdosteine also decreased the frequency of severe exacerbations, necessitating hospital admissions [217]. Erdosteine (300 mg two times a day) administration for 7–10 days ameliorated symptoms and decreased the length of hospitalization in individuals with exacerbations of COPD [221]. The administration of erdosteine (600 mg daily) with procysteine ameliorated the concentrations of various chemotactic cytokines including IL-6 and IL-8 and improved the concentration of CS-mediated ROS generation via alveolar macrophages in the bronchial secretions of tobacco smokers with COPD [220]. In COPD patients, erdosteine also reduced the concentrations of pro-inflammatory eicosanoids in the blood of patients with COPD [222].

### 8.2. Superoxide Dismutase Mimetics

There are three classes of SOD mimetics. The first class of SOD mimetics involves various manganese (Mn)-based macrocyclic ligands including M40419, M40403, and M40401 [113,223]. The second category involves multiple Mn-metaloporphyrins including AEOL-10150 and AEOL-10113 226,262], whereas the third class includes salens (Mn-based SOD mimetics). Since salens also have a catalase-like function, they can also neutralize ONOO^−^ and H_2_O_2_ [224]. In animal models of airway inflammation, so far, only the second category of SOD mimetics has been investigated. In rat models, AEOL10150 (a SOD mimetic) suppressed the tobacco smoke-mediated lung inflammation, LPO, and the production of ONOO^−^ [225]. It has been observed that treatment with recombinant SOD can reduce the cigarette smoke-induced release of IL-8 and prevent the influx of neutrophils into the lungs, which further indicates its potential as an anti-inflammatory and antioxidant in the case of COPD [95]. MnTE-2-PyP (a Mn porphyrin) is an antioxidant and ROS scavenger. MnTE-2-PyP has been reported to have the capacity to scavenge H_2_O_2_, ONOO^−^, lipid peroxides, and superoxide [226]. 

Treatment with MnTE-2-PyP decreased damage and inflammation-mediated via numerous factors [227,228]. The MnTE-2-PyP-mediated anti-inflammatory properties mainly take place because of its capacity to decrease MnTE-2-PyP. Collectively, these findings suggest that these compounds might have the potential to treat COPD. In the lungs, extracellular SOD (ECSOD or SOD3) is found to be highly expressed. ECSOD is found in the blood vessel linings of the lungs, the surface of the airway smooth muscle, and the extracellular matrix in the junctions of the airway epithelial cells [229]. SOD3 also has the capacity to directly scavenge O2^•−^, thus it has a significant contribution in providing protection against OS in the lung. Furthermore, SOD3 can reduce CS-mediated OS in mouse macrophages [230]. Moreover, SOD3 decreases emphysema and lung inflammation by reducing the oxidative fragmentation of the extracellular matrix (ECM) including elastin and heparin sulfate [231]. Thus, the discovery of therapeutic agents to increase or replenish the level of SOD3 in the lungs might be an effective therapeutic intervention for emphysema/COPD.

### 8.3. Nrf2 Activators

Nuclear factor erythroid-2-related factor 2 (Nrf2) is found in the cytoplasm of normal cells. This transcription factor has a significant contribution in providing protection against ROS and electrophiles. As a reaction to electrophilic stress and OS, Nrf2 detaches from Kelch-like ECH-associated protein 1 (Keap1) to translocate into the nucleus, where it has been reported to bind with the antioxidant response element (ARE) of target genes [232,233,234]. It has been observed that Nrf2 controls nearly all of the phase II cytoprotective genes and antioxidants, for instance, glutathione peroxidase (GPx), glutamate-cysteine synthase, glutamate-cysteine ligase modifier subunit (GCLM), NAD(P)H/quinone oxidoreductase 1 (NQO1), and multiple glutathione S-transferase family members [232]. Nrf2 knockout mouse models exhibited an elevated level of susceptibility to CS-mediated emphysema compared to wild-type mouse models, which suggests a protective function of Nrf2 [235,236]. 

DJ-1 (a Nrf2 stabilizer) loss and post-translational modification of the Keap1–Bach1 equilibrium can lead to Nrf2 downregulation in the lungs of COPD patients [237,238,239,240]. CDDO-imidazolide (a Nrf2 activator) was found to provide protection against CS-mediated emphysema in mouse models [234]. Nrf2 activation via sulforaphane (found in cruciferous vegetables and broccoli) can result in reducing some of the biochemical changes that take place in COPD patients and smokers [241]. Chalcones show anti-inflammatory properties because of their capacity to suppress the NF-κB signaling pathway [242,243] and activate the Nrf2/ARE signaling pathway therefore triggering expressions of phase II detoxifying enzymes [244]. Multiple chalcone derivatives are also being invented that are likely to have the potential to treat COPD [245]. Nevertheless, the toxicity, bioavailability, and pharmacokinetics of these compounds in the lungs are yet to be fully revealed.

### 8.4. NOX Inhibitors

In COPD, NOX (a membrane-bound complex) plays a role as an important source of ROS through the production of superoxide anions. There are various isoforms of the catalytic component of NOX such as dual oxidases (Duox1 and Duox2) and NOX1-5 [246]. Various inhibitors of NOX have also been invented to counter OS [247,248]. In CS-exposed mouse models, systemic administration of apocynin (a non-selective inhibitor of NOX) decreased the inflammatory chemokines and cytokines in BAL fluid [249]. Nebulized apocynin administration decreased the levels of nitrite and H_2_O_2_ in the exhaled air in patients with COPD, however, no clinical parameters were observed [250]. In addition to other functions, various polyphenols including resveratrol and quercetin suppressed the NOX function. Indeed, it is difficult to develop highly selective inhibitors of NOX. Setanaxib is a dual inhibitor of Nox1/4 that decreases acute lung damage mediated via reperfusion injury, however, this effect is yet to be confirmed in COPD models and clinical trials [251]. 

### 8.5. Antioxidant Mimetics

Antioxidant mimetics have been created to reinstate the depleted levels of endogenous antioxidants including GPx, catalase, and SOD [252]. SOD mimetics involve various metalloporphyrins including AEOL 10150, AEOL 10113, and Mn-containing molecules including M40419. These SOD mimetics were found to be effective in several animal models of OS, for example, CS-exposed mouse models showed a decreased level of inflammation [225]. AEOL 10150 is mainly used to treat radiation pneumonitis and the development of its derivatives is ongoing for the treatment of patients with COPD. GPx includes non-selenium and selenium-containing antioxidant enzymes that catalyze H_2_O_2_ breakdown. The level of GPx-1 is decreased in the plasma and lungs of COPD patients, which indicates that GPx-1 mimetics might be therapeutically beneficial [253]. GPx transgenic mouse models have protection against emphysema and inflammation following CS exposure, while GPx gene knockout elevated the lung response to CS [254]. Ebselen is a GPx mimetic and a selenium-based organic complex. This GPx mimetic is a strong antioxidant and it significantly neutralizes peroxynitrite radicals [255]. Ebselen suppressed the NF-κB/AP-1 activation, and therefore the expressions of pro-inflammatory genes in human leukocytes treated with peroxynitrite. Ebselen effectively decreased the ozone-induced airway inflammation in rat models [256] and inflammatory cytokines in the lungs of CS-exposed mouse models [249], however, no clinical trials have been carried out in COPD patients. 

### 8.6. iNOS Inhibitors and Spin Trapping Agents

It has been demonstrated that iNOS suppression via several chemical inhibitors including NG-nitro-L-arginine methyl ester and L-N6-(1-Iminoethyl)lysine (L-NIL) weakened animal models of pulmonary emphysema [257,258]. In addition to supplementation with other antioxidants, selective iNOS inhibition might be an effective approach in managing COPD [259]. Spin trapping agents are chemicals that can cause the quenching of free radicals to generate measurable stable end-products. Most of the spin trapping agents contain a nitroxide- or nitrone-nucleus and are derivatives of these moieties. Spin trapping agents have been extensively investigated in vitro. Their therapeutic potential has also been investigated in in vivo animal models of lung inflammation by utilizing α-phenyl-N-tert-butyl nitrone [260,261,262]. Spin trapping agents that were created earlier had very short half-lives and produced toxic hydroxyl radicals on decay. However, this problem has been solved by introducing electron-withdrawing moieties around the core of the pyrroline rings [263]. Azulenyl- and isoindole-based nitrones including azulenyl nitrone (STANZ) show potent antioxidant effects and have the capacity to suppress in vitro LPO [264,265]. On the other hand, derivatives of phenyl-base nitrone spin trap including NXY-059 exerted beneficial effects in several animal models of lung diseases [266,267,268].

### 8.7. Suppression of Nitrative Stress

Superoxide anions rapidly combine with NO to generate extremely reactive peroxynitrite ions, which can further lead to the generation of 3-nitrotyrosine adducts of amino acids in proteins that might disturb their activities as structural proteins, ion channels, or enzymes. The level of peroxynitrite is elevated in the EBC of patients with COPD [215], while 3-nitrotyrosine is expressed in the airways and sputum cells of COPD individuals [269,270]. The generation of peroxynitrite in the airways of patients with COPD might elucidate why the level of fractional exhaled NO (FeNO) is low in COPD individuals, since the entire free NO is closely bound by superoxide anions. NO is expressed in the alveolar epithelial cells. In response to OS, NO might be produced in the lungs of COPD patients by NOS1 [258]. Mouse models exposed to CS showed an elevated level in the expression of iNOS (NOS2) and were found to be protected against emphysema development by using selective inhibitors of iNOS and via knockout of the iNOS gene [271]. When aminoguanidine (a comparatively selective iNOS inhibitor) was administered through nebulization, it partly decreased peripheral and central exhaled levels of NO in COPD individuals, however, it failed to remove exhaled NO, indicating that NOS1 is the probable cause and those selective inhibitors of iNOS might not be beneficial in decreasing the level of peroxynitrite in patients with COPD [258]. Various highly selective inhibitors of iNOS have been invented for clinical administration. L-NIL (a selective inhibitor of iNOS) is highly effective in decreasing the level of FeNO in asthma patients; however, it is yet to be studied in patients with COPD [272].

### 8.8. Lazaroids and Edaravone

Lazaroids are a group of non-glucocorticoid analogs of methylprednisolone and they have the capacity to cross hydrophobic areas of the cell membrane, particularly to avert membrane LPO [261,273,274]. Their protective actions have already been demonstrated in numerous animal models of lung injury [274,275]. Lazaroids mainly exert their protective actions by suppressing LPO. Lazaroids suppressed the secretion of TNF-α from alveolar macrophages and the generation of free radicals [276,277]. However, more studies are needed to assess the effectiveness of lazaroids as a therapeutic agent in COPD treatment. Edaravone is an LPO inhibitor and a strong scavenger of protein carbonyls and free radicals [278,279]. Since carbonyl stress and protein carbonylation via aldehydes are present in COPD, edaravone can provide protection to the lungs against the activities of these oxidative products [280,281]. Moreover, edaravone improved the OS, inflammation, lung damage, and even mortality, mediated by intestinal ischemia-reperfusion in the rat models [282]. Collectively, all of these findings suggest that edaravone can be effective in COPD treatment.

### 8.9. Enzymatic and Small Molecule Antioxidants

Various antioxidant enzymes including GPx, catalase, and SOD can effectively neutralize cellular ROS. However, the functions and expressions of these antioxidant enzymes are changed in various diseases including OS. Reinstatement of altered antioxidant enzyme function can be attained via small molecules having catalytic effects, which can mimic the function of the enzyme. Various small molecule antioxidants have been created, however, only a few of them have been investigated in clinical studies. Various nitrone spin trap antioxidants including disufentan sodium were created as a neuroprotective agent, however it failed in clinical studies of acute stroke. Thioredoxin (Trx) (an endogenous regulator of redox balance) has the capacity to neutralize and its level might be decreased in COPD [283]. Systemic Trx-1 administration was found to be effective in reducing the neutrophilic inflammation in murine models of COPD [284]. Redox effector factor-1 (Ref-1) and Trx belong to the oxidoreductase family of redox sensors. Trx has the capacity to bind to various proteins including apoptosis signal-regulating kinase and hepatopoietin, which are secreted from these complexes during OS [285].

Following dissociation, Trx was found to reduce an important key thiol group within the p65/NF-κB subunit, which can further result in transcriptional activation [286,287,288]. In an animal model, suppression of Trx by MOL-294 (a small molecule Trx inhibitor) blocked the nuclear activation of AP-1-, and NF-κB-dependent transcription led to a reduced influx of neutrophils and the generation of TNF-α [289]. Trx activation through small molecules decreased the level of OS [290]. Trx-1 overexpression, mainly because of its antioxidant effect, decreased CS-induced OS and emphysema [291], but the actions in COPD have yet to be evaluated. It has been observed that XO might produce superoxide anions and is elevated in the lungs of mouse models exposed to CS [292]. An increased level in the expression of XO was observed in the bronchial mucosal lining fluid of patients with COPD and is linked with the elevated expressions of various proinflammatory cytokines [293]. Allopurinol (a XO inhibitor) (Figure 3) decreased the expression of 3-nitrotyrosine in the sputum cells of patients with COPD and elevated the level of FeNO, perhaps by blocking the superoxide production so that ONOO− is not generated [294].

### 8.10. Dietary Antioxidants

Diets that are poor in antioxidants have been linked to poor lung activity and can be a risk factor for COPD development, however, it is challenging to demonstrate that antioxidant vitamins precisely ameliorate established COPD [295]. There are various dietary antioxidants including flavonoids (e.g., quercetin), resveratrol (a dietary polyphenol), vitamin E, and vitamin C, however, improving the intake of dietary antioxidants has not been confirmed to ameliorate clinical features of COPD or lung activity [296,297]. It has been reported that the Mediterranean diet contains an increased level of dietary antioxidants and some evidence suggests that it might provide protection against COPD development; however, factors including poverty are challenging to separate out [298]. Resveratrol present in red wine and red-skinned fruits show in vitro anti-inflammatory and antioxidant properties in COPD cells and decreases lipopolysaccharide-induced pulmonary neutrophilic inflammation in rat models [299,300]. Nonetheless, its oral bioavailability is poor, therefore, stronger and more orally bioavailable analogs have been developed. Resveratrol, through inhaled formulations, decreased the enhanced lung aging in telomerase-deficient mice [301]. It has been reported that (-)-epigallocatechin (a polyphenol present in green tea) can cause the activation of FOXO3a, which is a transcription factor that controls various antioxidant genes including catalase and SOD [302].

### 8.11. Peroxidase Inhibitors

An increased level of neutrophil-derived MPO has been observed in the case of COPD, which reflects the activation of neutrophils in the lungs [303]. So far, multiple inhibitors of MPO have been created [304]. AZD5904 (a selective and irreversible inhibitor of MPO) decreases OS and emphysema development in CS-exposed guinea pigs [305]. Even though this drug was well-tolerated in human participants, it has been discontinued for unknown causes.

### 8.12. Mitochondria-Targeted Antioxidants

It has been confirmed that dysfunctional mitochondria are present in the case of COPD [306]. Mitochondrial mass was found to increase, along with the disruption and fusion of mitochondria with leakiness of the mitochondrial membranes [307]. This can be partially clarified by weakened autophagy processes that eradicate damaged mitochondria (mitophagy). In the case of COPD, dysfunctional leaky mitochondria might serve as the main source of ROS [308,309]. There are various available drugs that can particularly target mitochondria [310,311,312]. Mitochondria-targeted antioxidants have been developed primarily based on ubiquinone’s structure, which can be 50- to 100-fold concentrated in the mitochondrial matrix. Furthermore, they were found to be more potent than multiple animal models of aging [311,313]. A number of mitochondria-targeted antioxidants including SkQ1, pyrroloquinoline quinone, mito-TEMPO, and mitoQ are currently being studied in several clinical studies for age-linked diseases. mito-TEMPO also suppressed the mROS secretion and mitochondrial dysfunction induced by CS in the human airway epithelial cells in vitro [314]. Treatment with mitoQ also decreased lung inflammatory mediators, neutrophilic inflammation, and airway hyperresponsiveness in a murine model of the chronic OS that involved long-term exposure to ozone [308].

## 9. Strategies to Improve Pulmonary Bioavailability of Antioxidants

Indeed, the beneficial actions of antioxidants largely depend on their pharmacokinetic properties. Therefore, the bioavailability of antioxidants needs to be studied before assessing their antioxidant properties [315]. Poor pulmonary bioavailability has limited the use of antioxidants in COPD treatment [139,316]. There are various factors including host metabolism, release pattern from foods, and absorption, which can affect the bioavailability of exogenous and dietary antioxidants [315]. Host and dietary factors can markedly limit the half-life, concentration, and absorption of exogenous antioxidants. Therefore, it is important to address these factors to develop strong antioxidants that would effectively maintain constant concentrations within lung tissue compartments [315,317].

Modifying existing antioxidants or developing new antioxidants that are resistant to phase II modification would be able to retain their bioavailability and potency, even after modification via enzymes. Tempol, a novel membrane-permeable radical scavenger, is effective in decreasing lung inflammation in response to shock [318,319]. Ameliorated inhaled delivery approaches have been developed to avoid the first pass metabolism that takes place during systemic absorption and to mediate sufficient levels of antioxidants to be reached in the lung [320,321]. Currently available inhalational devices are able to deliver most of the drugs only to medium and large size airways of the lung, while ignoring the alveolar regions and small airways that are major areas in the disease pathogenesis [322]. In addition, ameliorations in distal lung deposition are necessary before utilizing this technology for the effective delivery of antioxidants. In the near future, this might be achievable, since there are effective inhalational techniques that can mediate uniform distribution in the distal lung [323,324]. Future improvements in antioxidant therapy might include counteracting the CS-mediated stimulation of various oxidant-forming enzymes including xanthine oxidase and NOX, which are found within the lung epithelium. Moreover, these enzymes can play significant roles in OS, which is mediated by exposure to CS [325,326,327,328].

## 10. Future Directions

There is growing evidence that suggests that elevated ROS generation takes place in the case of COPD, which might be crucial for COPD pathogenesis. Various small molecules are currently being investigated in clinical studies that quench oxidants or target oxidant signaling induced by CS. In order to address these issues of COPD, various anti-inflammatory agents and/or antioxidants including antioxidant mimetics, dietary polyphenols, spin trapping agents, thiol molecules, and inhibitors of OS-mediated signaling pathways have been developed. In addition, antioxidants might increase the effectiveness of glucocorticoids by quenching aldehydes and oxidants, which further increases the histone deacetylase function in patients with COPD. Various dietary polyphenols including curcumin and resveratrol can also suppress the release of proinflammatory cytokines, histone acetylation, oxidant/CS-induced activation of NF-κB, and restore the functions of glucocorticoids by upregulating the function of histone deacetylase. Therefore, dietary polyphenols control inflammation at the molecular level, which might further restore the functions of glucocorticoids in the treatment of CS-mediated chronic inflammatory diseases. Thus, a potent and effective antioxidant with good bioavailability is important to regulate inflammatory and localized oxidative mechanisms that are present in COPD pathogenesis [329].

## 11. Conclusions

Indeed, OS and carbonyl stress play important roles in COPD pathogenesis. Targeting OS with potent antioxidants or increasing the endogenous antioxidants may be beneficial in the treatment of COPD. Antioxidants might also affect various issues of COPD including ECM remodeling, inflammation, the hypersecretion of mucus, and overcoming steroid resistance. Various small molecule antioxidants have been evaluated in preclinical and clinical studies. Even though thiol-based antioxidants can effectively target the ROS and cellular responses induced by oxidants, novel and more potent antioxidants with good bioavailability that can be used in COPD treatment need to be studied in clinical studies. Nevertheless, a limited number of clinical trials have been conducted and there is a deficiency of data regarding the toxicity, bioavailability, pharmacokinetics, and absorption of multiple activators of endogenous antioxidants and exogenous antioxidants.

## Figures and Tables

**Figure 1 molecules-27-05542-f001:**
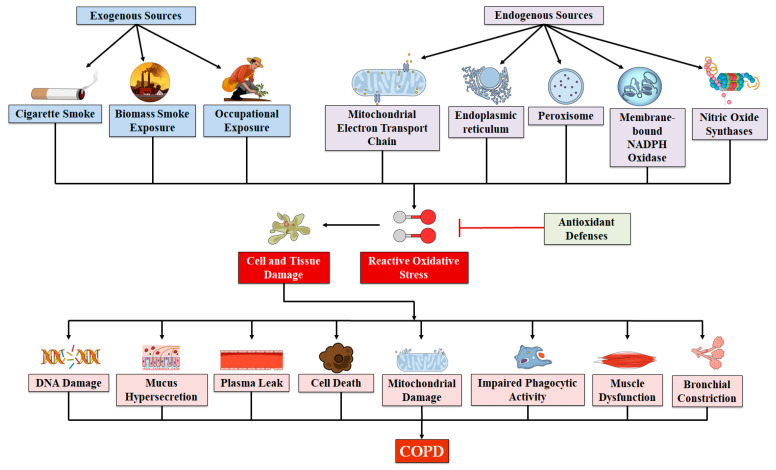
A schematic presentation of the role of oxidative stress in the development of COPD.

**Figure 2 molecules-27-05542-f002:**
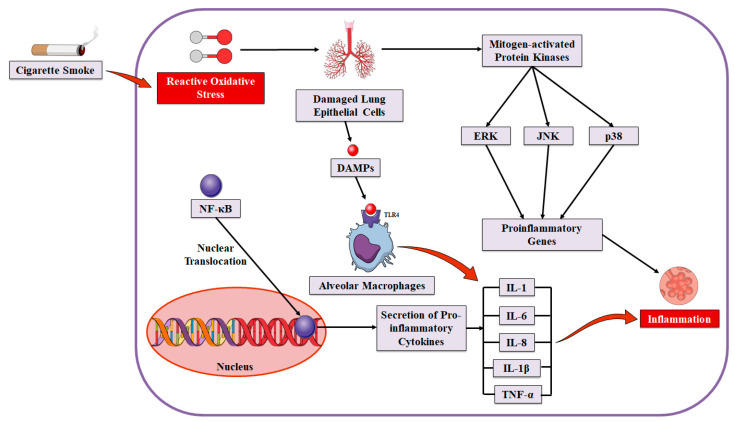
The effects of cigarette smoking on oxidative stress and inflammation in alveolar epithelial cells. Abbreviations: ERK—extracellular signal-regulated kinase; IL-1—interleukin 1; IL-1β—interleukin-1β; IL-6—interleukin 6; IL-8—interleukin 8; JNK—jun-N-terminal kinase; NF-κB,—nuclear factor-κB; TNF-α—tumor necrosis factor α.

**Figure 3 molecules-27-05542-f003:**
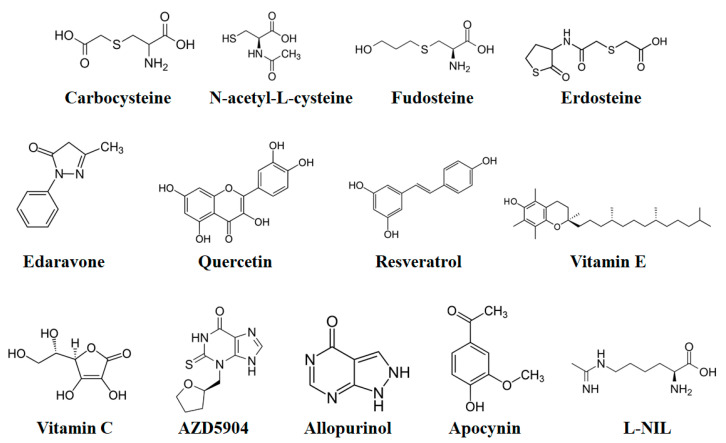
The chemical structures of various antioxidants that can be used to target oxidative stress in the treatment of COPD.

**Table 1 molecules-27-05542-t001:** The risk factors associated with the development of chronic obstructive pulmonary disease.

Risk Factors		References
External	Smoking	[19,20]
	Biomass smoke exposure	[21,22]
	Low socioeconomic status	[23,24,25]
	Occupational exposures	[26,27]
Internal	Alpha-1-antitrypsin deficiency	[28,29]
	Gender differences	[30,31,32,33]
	Airway mucus hypersecretion	[34,35]
Other	Airway hyperresponsiveness	[36,37]
	Early life insults	[38,39]
	Air pollution	[40,41,42]
	Asthma	[43,44,45]
	Malnutrition	[46]

**Table 3 molecules-27-05542-t003:** The clinical studies on the efficacy of carbocysteine in COPD.

Antioxidant	Study Design	Duration	Study Outcomes	References
Carbocysteine	Double-blind, randomized, placebo-controlled study	12 months	Prolonged (12 months) treatment with carbocysteine decreased the exacerbations in COPD patients, reduced exacerbations, no loss of lung activity, ameliorated health-related quality of life.	[187,192]
Carbocysteine	Multicenter, placebo-controlled, double-blind, parallel group trial	6 months	Duration of the acute respiratory illness was markedly decreased and this was linked with a marked decrease in the administration of antibiotics during the trial period, no serious adverse effects were observed.	[193]
Carbocysteine	Randomized controlled trial	12 months	Consistently decreased the frequency of exacerbations, did not alter the lung function.	[188]
Carbocysteine	Double-blind controlled study	3 months	Improved the capacity to cough up bronchial secretions, markedly elevated the sputum volume output, ameliorated ventilation.	[194]
Carbocysteine	Single-blind study	8 weeks	Greatly eased expectoration, increased the level of expectorated sputum, markedly increased peak expiratory flow rate, ameliorated the severity of dyspnea.	[195]
Carbocysteine	Randomized controlled trial	12 months	Markedly decreased the exacerbation rate and commoncolds, no substantial differences in the extent of COPD.	[196,197]

**Table 4 molecules-27-05542-t004:** The clinical studies on the efficacy of N-acetyl-L-cysteine in COPD.

Antioxidant	Study Design	Duration	Study Outcomes	References
N-acetyl-L-cysteine (NAC)	Randomized, placebo-controlled trial	3 years	NAC is not effective at preventing deterioration of lung activity and exacerbations in COPD patients.	[202,203]
NAC	Double-blind, double dummy, randomized comparison study	12 months	Long-term oral administration of NAC reduces H_2_O_2_ generation in the airways of patients with COPD.	[204]
NAC	Double-blind, randomized, placebo controlled trial	7 days	NAC introduction in the treatment with bronchodilators and corticosteroids does not alter the outcome in acute exacerbation of COPD.	[205]
NAC	Randomized, controlled trial	12–24 weeks	Prevention of exacerbation and improved symptoms as compared to 34.6% of participants receiving a placebo.	[206]
NAC	Single-blinded, randomized trial	2 months	Oral administration of NAC for 2 months quickly decreases the oxidative stress in the airways of patients with COPD.	[207]
NAC	Randomized, controlled trial	2 months	No significant alteration in lung activity was observed; marked decrease in the duration of disability and a 29% decrease in exacerbations.	[208]
NAC	-	5 days	Increased levels of glutathione and cysteine on day 5.	[209]

## Data Availability

Not applicable.

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
