# Peer review of "Therapeutic Potential of Small Molecules Targeting Oxidative Stress in the Treatment of Chronic Obstructive Pulmonary Disease (COPD): A Comprehensive Review"

_molecules, 2022, doi:10.3390/molecules27175542_

Round 1

Reviewer 1 Report

Thank you for inviting me to review the article titled “Therapeutic Potential of Small Molecules Targeting Oxidative Stress in the Treatment of Chronic Obstructive Pulmonary Disease (COPD): a Comprehensive Review.”

This article details the critical role of oxidative stress induced by cigarettes, among others, in the pathogenesis of COPD. It is proposed that small molecules, thiol-based antioxidants, modulators of protein carbonylation and lipid peroxidation, antioxidative enzymatic mimetics, and spin trapping agents were selected according to their therapeutic potential for COPD control. This article summarizes the role of oxidative stress in the pathogenesis of COPD and focuses on the therapeutic potential for 11 classes of antioxidants in treating COPD.

Overall, the article is informatively organized with fluent expression and logical coherence. Two minor revisions need to be considered before publication:

1. The authors summarize that 11 small molecule compounds treat COPD by modulating oxidative stress. Is there a certain intrinsic relationship between the chemical structures (e.g., functional groups) of the 11 molecules and their antioxidant /or anti-inflammatory potentials? As we know, the credible drug ability was established on various characteristics in silico, in vitro, and in vivo. For example, resveratrol (trans -3,5,4′-trihydroxystilbene) has been proved to be an excellent anti-inflammatory drug in vitro with an extremely short half-life in vivo. Further, quercetin, a common flavonoid, has low bioavailability, which limits its oral application. Therefore, structure clues should be more evident towards both antioxidative activities and utilization. Thus, the QSAR study and ADME properties were essential to understanding the COPD drug better.

2. The oxidative stress (OS) is a complex process that refers to a series of intracellular reactions among oxides, enzymes, and pathways, which happen at a large space-time scale. The pathological inducers of COPD caused not only OS, but also led to neutrophil inflammatory infiltration, increased protease secretion, and other secondary metabolites. there are few descriptions of the specific mechanisms or targets in which small molecule compounds interfere with oxidative stress in this article. So, it is hard to limit the pharmacological actions or results in a broad spectrum of those therapeutic agents precisely on “Targeting”. In addition, the article does not concern with intervention times for controlling COPD, e.g., pre-, mid-, and post-prom COPD onset, which may mislead the strategy for preventive or therapeutic issues.

Author Response

Reviewer 1

  1. The authors summarize that 11 small molecule compounds treat COPD by modulating oxidative stress. Is there a certain intrinsic relationship between the chemical structures (e.g., functional groups) of the 11 molecules and their antioxidant /or anti-inflammatory potentials? As we know, the credible drug ability was established on various characteristicsin silicoin vitro, and in vivo. For example, resveratrol (trans -3,5,4′-trihydroxystilbene) has been proved to be an excellent anti-inflammatory drug in vitrowith an extremely short half-life in vivo. Further, quercetin, a common flavonoid, has low bioavailability, which limits its oral application. Therefore, structure clues should be more evident towards both antioxidative activities and utilization. Thus, the QSAR study and ADME properties were essential to understanding the COPD drug better.

Reply: QSAR or any other computational modeling methods is not the focus of my manuscript. Rather I have summarized the findings from clinical trials, in vitro and in vivo studies and highlighted the role of OS in COPD pathogenesis, and the therapeutic potential of numerous antioxidants in the treatment of COPD.

However, I have added an entirely new section regarding the strategies to improve the pulmonary bioavailability of antioxidants, which can be found in blue-colored texts in the revised manuscript.

  1. The oxidative stress (OS) is a complex process that refers to a series of intracellular reactions among oxides, enzymes, and pathways, which happen at a large space-time scale. The pathological inducers of COPD caused not only OS, but also led to neutrophil inflammatory infiltration, increased protease secretion, and other secondary metabolites. There are few descriptions of the specific mechanisms or targets in which small molecule compounds interfere with oxidative stress in this article. So, it is hard to limit the pharmacological actions or results in a broad spectrum of those therapeutic agents precisely on “Targeting”. In addition, the article does not concern with intervention times for controlling COPD, e.g., pre-, mid-, and post-prom COPD onset, which may mislead the strategy for preventive or therapeutic issues.

Reply: There are an enormous number of studies and evidence that have already indicated that oxidative stress (OS) is the key role player in COPD pathogenesis. The major initiating and risk factors of COPD including smoking and air pollution lead to OS. Moreover, it has already been confirmed by a large number of studies that elevated levels of mucus secretion, protease activity, and transcription of inflammatory genes are the consequences of the increased level of OS. Therefore, these events could be controlled by controlling OS. In the case of COPD, these events are the consequences of OS, rather than causes.

Along with antioxidant properties, a number of antioxidants also show anti-inflammatory and antibacterial effects. In this manuscript, I have already extensively covered that OS is present in various stages of COPD, therefore the use of antioxidants might be useful in controlling those OS events. It has also been reported that (reference: https://www.ncbi.nlm.nih.gov/pmc/articles/PMC1781748/) conventional COPD treatments including bronchodilators and inhaled steroids are often ineffective in combating OS present in the lungs of COPD patients. Therefore, the use of antioxidants might be a logical therapeutic agent in COPD treatment.

In the case of all the antioxidants mentioned in the manuscript, I have provided the specific OS inhibition mechanism of antioxidants, for example “N-acetyl-L-cysteine neutralizes oxidant species by reducing disulfide bonds and increase the level of intracellular GSH, Fudosteine acts as an antioxidant via elevating the level of intracellular cysteine, recombinant SOD can reduce cigarette smoke-induced release of IL-8 and prevent the influx of neutrophils into the lungs, which is further suggesting its potential as an anti-inflammatory and antioxidant in the case of COPD …..etc”. 

Reviewer 2 Report

In the manuscript, the authors reviewed the important role of OS in the pathogenesis of COPD and expounded the potential of multiple antioxidants in the treatment of COPD. This study has certain reference value for studying the etiology of COPD and researching therapeutic drugs, but there are some key issues and content that need to be discussed and supplied.

Comment 1: In the introduction, author wrote that COPD is the third leading cause of death worldwide as per the Global Burden of Disease 40 (GBD), the reference which can prove authors opinion was published in 2012, many years have passed and the data will change, authors should refer to recently published articles.

Comment 2: Most of the references referred by author are before 2020. We suggested that the author appropriately increase the reports in recent 5 years as references.

Author Response

Reviewer 2

Comment 1: In the introduction, author wrote that “COPD is the third leading cause of death worldwide as per the Global Burden of Disease 40 (GBD)”, the reference which can prove authors’ opinion was published in 2012, many years have passed and the data will change, authors should refer to recently published articles.

Reply: I have updated the text and provided more recent information and reference in the revised manuscript, which can be found in blue-colored texts.  

Comment 2: Most of the references referred by author are before 2020. We suggested that the author appropriately increase the reports in recent 5 years as references.

Reply: The references that are before 2020 contain information regarding established facts of the relevant topics. However, I have added more information from sources published in the last 5 years, which can be found in blue-colored texts.